# Permanent Magnet Synchronous Motor Control Based on Phase Current Reconstruction

**Guozhong Yao, Yun Yang, Zhengjiang Wang * and Yuhan Xiao**

Faculty of Transportation Engineering, Kunming University of Science and Technology, Kunming 650504, China
* Correspondence: wangzhengjiang@kust.edu.cn; Tel.: +86-158-7792-4502

**Abstract:** The traditional single current sensor control strategy of a permanent magnet synchronous motor (PMSM) often adopts the DC bus method, which makes it difficult to eliminate the blind area of current reconstruction. Therefore, a current reconstruction method based on a sliding mode observer is proposed. Based on the current equation of the motor, the method takes the α-axis and β-axis currents as the observation objects and shares the same synovial surface, so that the α-axis current observation value and the β-axis current observation value converge to the actual current value at the same time and the unknown β-axis current information is obtained. The control system first tests the performance of the motor under different working conditions when the parameters are matched, and then tests the current reconstruction ability of the parameter mismatch. The results show that the current observer with a matched parameter can accurately and quickly reconstruct the β-axis current under various operating conditions, and the maximum current error does not exceed 4 mA. When the parameters are mismatched, high-performance control of the motor can still be achieved. The proposed method has excellent robustness.

**Keywords:** PMSM; single current sensor; current reconstruction; sliding mode control; robustness

## 1. Introduction

A permanent magnet synchronous motor (PMSM) has the advantages of high power density and fast dynamic response. It is widely used in industrial production, new energy vehicles, ship propulsion, and other fields [1–4].

The control of the motor usually uses constant volts per hertz (V/f), direct torque control (DTC), field-oriented control (FOC), and other methods. V/f control has matured for a long time [5]. It can generate the required torque by coordinating the amplitude and frequency of the stator voltage, which is the simplest motor control strategy. However, the performance of V/f control under low frequency and high load conditions is far less than that of other control methods [6], which leads to the limitation of V/f control applications in situations where transient performance is high [7], and high-performance motor control often uses other methods. Compared with FOC, DTC is less dependent on rotor position [8], but has larger torque and flux fluctuations [9–11]. The research on sensorless FOC has a history spanning more than 30 years [12–14]. The dependence on the rotor position cannot limit the development of FOC. FOC control has become the mainstream control method for the motor. Current feedback is an indispensable part of FOC. The traditional FOC control strategy often uses two or three current sensors to obtain current information. The single current sensor motor has received extensive attention due to its small system size and low production cost [15–18].

The traditional PMSM single current control strategy generally adopts two methods: the DC bus method and the current observer method. As early as 1989, the DC bus method was proposed [19]; its essence is to use the instantaneous current of the inverter fixed point, which contains multi-phase current information, to reconstruct the current. However, it takes time to collect current information, and the effective voltage vector

has a short action time, which leads to the existence of the current reconstruction blind area. In recent decades, many improvement strategies have been proposed to overcome the current blind spot. Reference [20] proposed a three-state pulse width modulation technology that divides the space vector into two regions. In the low modulation ratio region, two basic voltages with a difference of 120° are used to synthesize the reference voltage. This technology reduces the blind area of current reconstruction and suppresses the common mode voltage, but its performance is not as good as the traditional seven-segment space vector pulse width modulation control method. In reference [21], it is proposed to use an isolated current sensor instead of a DC bus current sensor to sample twice in each PWM cycle. However, this method requires additional leads, which will introduce parasitic parameters and lead to system performance degradation. At the same time, the current reconstruction blind area in the overmodulation region is too large, which is also an important factor limiting the method. The PWM phase shift method proposed in reference [22] can increase the observable area of the current by moving the PWM waveform when the non-zero voltage vector action time is too short, but this method still has a voltage vector action time after phase shift that is less than the minimum sampling time. Reference [23] proposed a new phase current reconstruction technique in which a single current sensor is installed on a branch rather than a DC bus. The blind area of the current reconstruction in the sector boundary region and the low modulation region can be reduced without introducing any additional algorithm compensation strategy, but there is a blind area in the high modulation ratio region. Reference [24] proposed a new phase current reconstruction scheme without using a zero-switching state. The current reconstruction dead zone is divided into six sectors. Each sector is divided into three parts by the corresponding vector synthesis method to obtain enough effective switching states, thereby reducing the current reconstruction dead zone, but it will increase the current ripple and switching loss and reduce the service life of the system. Reference [25] proposed the idea of "substitution." When the reference voltage vector is located in the blind area of overmodulation reconstruction, it is replaced by the adjacent reconfigurable maximum voltage vector to broaden the operating range of the phase current reconstruction technology, but this does not solve the blind area problem in other areas.

The core idea of the above improvement measures is to be able to collect the required current information in time by changing the space voltage vector or switching states within a limited time. However, due to factors such as the high dynamic response of motor control, the requirements of space voltage vectors and switching are very high. Once these measures are adopted, it is not only difficult to completely eliminate the blind area of the current reconstruction, but it also has a negative impact on the control performance of the motor. Therefore, the current observer becomes another direction in the current reconstruction. The current observer method without a low modulation ratio region and a sector transition region has certain advantages.

In [26], an adaptive observer is proposed to realize the single current control of the motor. The observer equation and adaptive law are not complicated, but their robustness depends on parameter identification. Incorrect parameter estimation will bring errors to current reconstruction. The ESO observer designed in reference [27] can realize the accurate control of a single current sensor, but the parameter design of ESO is very difficult. It is usually obtained by the trial-and-error method, which not only increases the workload of the algorithm but also makes it difficult to obtain the optimal parameters. Verma et al. [28] reconstructed the β-axis current based on the mathematical model of the motor and the PI loop, but the anti-interference ability of the PI loop needs to be improved. Reference [29] designed a single current sensor control algorithm based on Kalman filter, but the calculation of the correction current is not accurate enough, the control performance is poor, the motor torque accuracy is low, and there is a non-periodic pulse spike. Reference [30] can realize single current sensor control without a position sensor, but the robustness of the system to the stator resistance is not very good. Reference [31] combines the DC bus method and the

observer method and uses the Luenberger observer to compensate for the limitation of the sector boundary region, but there is still a blind area in the low modulation region.

The above current observers all introduce complex structures, which increase the computational burden and have a certain impact on the real-time performance of motor control. In addition, the error of current reconstruction and external interference cannot achieve the ideal current reconstruction effect. Table 1 briefly analyzes the current reconstruction strategy.

**Table 1.** Brief comparisons of single current control strategies.

| Method | References | Advantages | Disadvantages |
|---|---|---|---|
| DC bus method | Reference [19] | The DC bus method is proposed. | The current blind area. |
| | Reference [20] | Reduces the blind area and the common-mode voltage. | Not as good as the traditional 7-segment SVPWM. |
| | Reference [21] | Needs an isolated current sensor. | Additional lead wire. |
| | Reference [22] | PWM Phase Shift. | Cannot eliminate all blind area. |
| | Reference [23] | One Branch. | A new blind area. |
| | Reference [24] | Switching state phase shift method. | Current ripple and switch damage. |
| | Reference [25] | Substitution method. | Ignore the sector boundary. |
| Current observer method | Reference [26] | Observer equations and adaptive laws are not complicated. | Robustness depends on parameter identification. |
| | Reference [27] | Accurate. | ESO parameters. |
| | Reference [28] | Based on the PI loop. | PI ring is not robust. |
| | Reference [29] | EKF. | Low accuracy. |
| | Reference [30] | The estimation technique is independent of machine parameters. | The robustness of stator resistance is not very good. |
| | Reference [31] | The DC bus method and observer method are combined. | A blind area in the low modulation region. |

At present, regarding the PMSM single current sensor control system, there is no system that can completely eliminate the blind area of current reconstruction and has the characteristics of a simple structure and strong robustness. The elimination of the blind area in the current reconstruction will inevitably make the motor run more smoothly. Without introducing complex structure, the anti-interference ability is greatly enhanced, the real-time control effect is better, and the motor can adapt to a more complex working environment. Therefore, the purpose of this paper is to first realize the current reconstruction with no blind area, high precision, and good robustness by a very simple method, and then realize the high performance control of PMSM by a single current sensor.

## 2. Mathematical Model of PMSM

The control strategy is based on the α-β two-phase stationary coordinate system. The voltage equation of PMSM can be expressed [32] as follows:

$$\begin{bmatrix} u_\alpha \\ u_\beta \end{bmatrix} = \begin{bmatrix} R_S + pL_d & \omega_e(L_d - L_q) \\ -\omega_e(L_d - L_q) & R_S + pL_q \end{bmatrix} \begin{bmatrix} i_\alpha \\ i_\beta \end{bmatrix} + \begin{bmatrix} E_\alpha \\ E_\beta \end{bmatrix} \tag{1}$$

Among them, $u_\alpha$ and $u_\beta$ represent the voltage component of the α-β two-phase stationary coordinate system, where $R_S$ is the stator resistance and $\omega_e$ is the angular velocity of the motor. $L_d$ and $L_q$ are the inductance components on the two-phase rotating coordinate axis. $i_\alpha$ and $i_\beta$ are the current components on the α axis and the β axis, respectively, and $E_\alpha$ and $E_\beta$ represent the extended back electromotive force.

The voltage equation of Equation (1) can be written as the current equation:

$$\frac{d}{dt}\begin{bmatrix} i_\alpha \\ i_\beta \end{bmatrix} = \frac{1}{L_d}\begin{bmatrix} -R_S & -\omega_e(L_d - L_q) \\ \omega_e(L_d - L_q) & -R_S \end{bmatrix}\begin{bmatrix} i_\alpha \\ i_\beta \end{bmatrix} + \begin{bmatrix} u_\alpha \\ u_\beta \end{bmatrix} - \frac{1}{L_d}\begin{bmatrix} E_\alpha \\ E_\beta \end{bmatrix} \tag{2}$$

$\psi_f$ is the stator flux, and the extended back electromotive force calculation equation can be expressed as follows:

$$\begin{bmatrix} E_\alpha \\ E_\beta \end{bmatrix} = \begin{bmatrix} -\psi_f \omega_e sin\theta \\ \psi_f \omega_e cos\theta \end{bmatrix} \tag{3}$$

## 3. Current Reconstruction

### 3.1. Current Sliding Mode Observer Design

Chakraborty [33] proposed that in the dual current sensor control system, if the a-phase current is measured in error, the α-axis and β-axis currents will be wrong; if only the b-phase current is measured incorrectly, the β-axis current will be wrong, while the α-axis current is still correct.

Equation (4) is obtained by the Clark transformation, and Kirchhoff's current law fully shows that the α-axis current is only related to the a-phase current, while the β-axis current is related to the a and b two-phase currents. Obviously, if the single current sensor measures the b-phase current, the current of the α-axis and β-axis cannot be obtained; if the single current sensor measures the a-phase current, the α-axis current will be measurable, and the current that needs to be reconstructed is only the β-axis current. The three-phase reconstruction current can be obtained by a further inverse Clark transformation.

$$
\begin{bmatrix} i_\alpha \\ i_\beta \end{bmatrix} = \begin{bmatrix} 1 & 0 \\ \frac{\sqrt{3}}{3} & \frac{2\sqrt{3}}{3} \end{bmatrix} \begin{bmatrix} i_A \\ i_B \end{bmatrix}
\tag{4}
$$

The traditional sliding mode observer knows the current and voltage information in the stationary two-phase coordinate system. The purpose is to obtain the extended back electromotive force information, that is, so that Equation (3) can further analyze the rotor position and speed information. The traditional sliding mode observer can be expressed by (2) as follows:

$$
\frac{d}{dt} \begin{bmatrix} \hat{i}_\alpha \\ \hat{i}_\beta \end{bmatrix} = \frac{1}{L_d} \begin{bmatrix} -R_S & -\omega_e(L_d - L_q) \\ \omega_e(L_d - L_q) & -R_S \end{bmatrix} \begin{bmatrix} \hat{i}_\alpha \\ \hat{i}_\beta \end{bmatrix} + \frac{1}{L_d} \begin{bmatrix} u_\alpha - ksgn(\hat{i}_\alpha - i_\alpha) \\ u_\beta - ksgn(\hat{i}_\beta - i_\beta) \end{bmatrix}
\tag{5}
$$

In Equation (5), considering that the α-axis current is obtained indirectly by the a-phase current sensor, while the β-axis actual current is unknown, and the β-axis actual current is only applied in the sliding mode surface $(\hat{i}_\beta - i_\beta)$, if it is replaced by other known information and the sliding mode observer can still operate normally, then the β-axis current will be reconstructed, that is as follows:

$$
\begin{bmatrix} -R_S & -\omega_e(L_d - L_q) \\ \omega_e(L_d - L_q) & -R_S \end{bmatrix} \begin{bmatrix} \hat{i}_\alpha \\ \hat{i}_\beta \end{bmatrix} + \frac{1}{L_d} \begin{bmatrix} u_\alpha - ksgn(\hat{i}_\alpha - i_\alpha) \\ u_\beta - ksgn(\hat{i}_\alpha - i_\alpha) \end{bmatrix}
\tag{6}
$$

However, if according to this design, the α-axis observation current of the sliding mode observer still converges normally to the current value measured by the sensor, but the β-axis current is completely out of control. Considering that the β-axis current does not converge to the actual measurement value due to the lack of β-axis current error information $(\hat{i}_\beta - i_\beta)$, in the traditional sliding mode observer, in order to obtain the back electromotive force $E_\beta$, $ksgn(\hat{i}_\alpha - i_\alpha)$ is used instead of $E_\beta$ in the sliding mode observer. When the sliding mode observer converges, the two are equal. In fact, $E_\beta$ also contains the β-axis current error information $(\hat{i}_\beta - i_\beta)$. Therefore, on the basis of Equation (5), the following observer can be designed by introducing the back electromotive force information into the current equation of the motor as follows:

$$
L_d \dot{\hat{i}}_s = \begin{bmatrix} -R_S & -\omega_e(L_d - L_q) \\ \omega_e(L_d - L_q) & -R_S \end{bmatrix} \hat{i}_s + \begin{bmatrix} u_\alpha - E_\alpha - q \cdot sigmoid(\tilde{i}) \\ u_\beta - E_\beta - t \cdot sigmoid(\tilde{i}) \end{bmatrix}
\tag{7}
$$

where q and t are constants, $\tilde{i} = \hat{i}_\alpha - i_\alpha$, $\hat{i}_s = \begin{bmatrix} \hat{i}_\alpha \\ \hat{i}_\beta \end{bmatrix}$

The current observer mainly has the following characteristics:

(1)　α-axis and β-axis share the same sliding surface, namely $sigmoid(\tilde{i})$. This is due to the lack of important information on the traditional β-axis sliding mode surface, namely

the actual value of the β-axis current. After introducing the back-EMF containing the β-axis information, the α-axis current observation value and the β-axis current observation value will converge to the actual value at the same time, so the same sliding mode surface can be used for observation.

(2)   This design reconstructs the β-axis current instead of obtaining the back electromotive force. The reconstruction of the current requires the back electromotive force. If the acquisition method of the electromotive force is non-inductive, then it will be a single current sensing PMSM control strategy without a position sensor.

(3)   The quasi-sliding function $sigmoid(\tilde{i})$ is used to replace the traditional symbol function $sgn(\tilde{i})$. The smooth, continuous characteristic is more stable than the step characteristic, which can improve the performance of the sliding mode observer.

### 3.2. Sliding Mode Convergence Verification

Whether the sliding mode observer meets the expectation depends on whether it can converge to the sliding mode surface. The sliding mode convergence is proved below, the sliding mode surface $s = \hat{i}_\alpha - i_\alpha$ is defined, and Equation (8) is selected as the candidate function according to Lyapunov theory [34].

$$F(s) = \frac{s^2}{2} \tag{8}$$

The sliding mode observer is proved to be stable by satisfying the following conditions.

$$F(s) = \frac{s^2}{2} > 0 \tag{9}$$

$$\dot{F}(s) = s \cdot \dot{s} < 0, (s \neq 0) \tag{10}$$

The α-axis current observer value is infinitely close to the actual value, fluctuating around the actual value, but it will not be equal to the actual value, so s will not be equal to 0. The condition (9) is obviously established, and then whether the synovial membrane converges or not can be judged by whether the condition (10) is true.

By taking (4) minus (1), we can obtain the following:

$$\dot{s} = \frac{1}{L_d} \begin{bmatrix} R_S + pL_d & \omega_e(L_d - L_q) \\ -\omega_e(L_d - L_q) & R_S + pL_q \end{bmatrix} \tilde{i} - \frac{1}{L_d} sigmoid(\tilde{i}) \tag{11}$$

If $s > 0$, then $\dot{s} < 0$ satisfies the condition that follows:

$$-R_S|s| - \omega_e(L_d - L_q)\tilde{i}_\beta - qsigmoid(s) < 0 \tag{12}$$

In the surface-mounted permanent magnet synchronous motor, if the d-axis and q-axis inductance components are equal, and the constant q is designed to be positive, then the Equation (12) holds.

If $s < 0$, then $\dot{s} > 0$ satisfies the condition that:

$$R_S|s| - \omega_e(L_d - L_q)\tilde{i}_\beta - qsigmoid(s) > 0 \tag{13}$$

Similarly, if the constant q is designed to be positive, then Equation (13) holds.

In summary, when *q* and *t* are positive, the sliding mode will converge, but the specific parameters need to be adjusted according to the different motors to reduce the error. Two different parameters are set to adjust the current error, and the two parameters can be equal.

### 4. Experimental Verification

Through the above analysis, the control system shown in Figure 1 is built to verify the effect of the β-axis current reconstruction. The motor parameters are shown in Table 2.

Based on the control system of the current observer, the a-phase current is collected to reconstruct the three-phase current for current feedback, and the high-performance motor control of a single current sensor is realized.

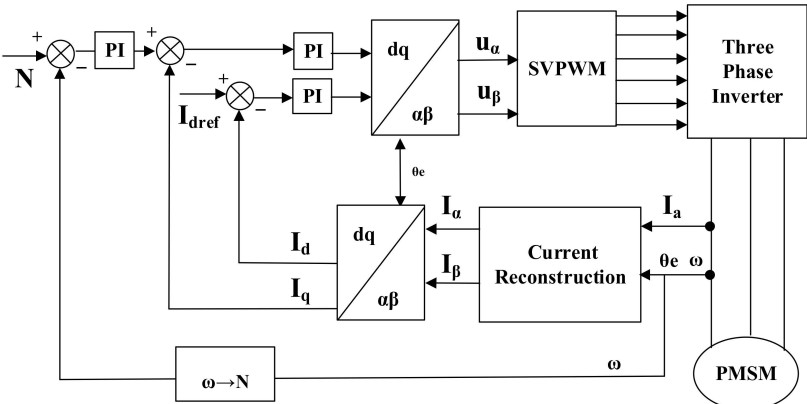

**Figure 1.** Control block diagram.

**Table 2.** System Parameters.

| Parameters | Value | Unit |
|---|---|---|
| Polar logarithm | 4 | – |
| Stator resistance | 2.875 | $\Omega$ |
| Stator inductance | 8.5 | mH |
| Flux linkage | 0.175 | Wb |
| Moment of inertia | 0.001 | Kg·m$^2$ |
| PWM frequency | 10 | kHz |

### 4.1. Analysis of Steady-State and Dynamic Reconstructed Current under Matched Parameter

The working condition W is no-load at a speed of 1000 rpm. According to the a-phase current and rotor position information, the sliding mode current observer is used to reconstruct the β-axis current, as shown in Figures 2 and 3, which show the error between the reconstructed β-axis current and the actual β-axis current. The reconstructed β-axis current is almost consistent with the measured current of the sensor. After the start-up phase, the β-axis current error is periodic. Most of the errors are concentrated within ±2 mA, and a few errors exceed the fluctuation range of ±2 mA. The maximum current error is only 4 mA in the start-up phase. After entering the stable phase, the maximum current error is reduced to 3 mA, and the maximum current error is only 0.85% of the steady-state current amplitude.

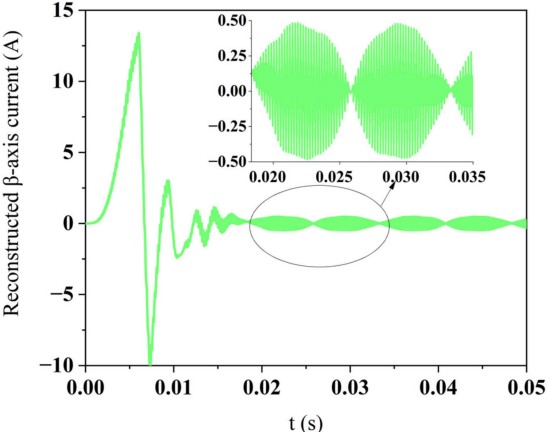

**Figure 2.** Reconstructed β-axis current.

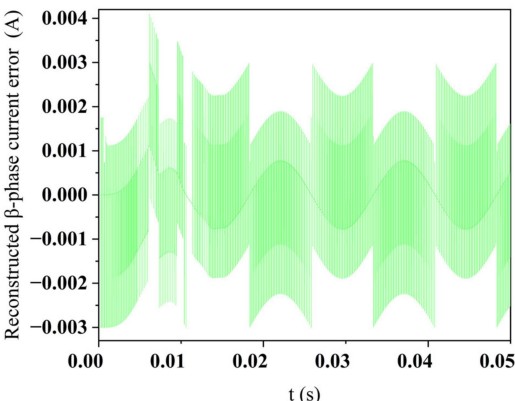

**Figure 3.** Reconstructed β-axis current error.

The motor control system shown in Figure 1 aims to replace two or three current sensors with a single current sensor, so it is necessary to analyze the error of reconstructing three-phase current. The a-phase current is collected by the sensor, and the reconstruction of the b- and c-phase currents requires the participation of the β-axis current. The β-axis current must have a reconstruction error, so that the reconstructed b- and c-phase currents must also have errors, as shown in Figures 4 and 5. Because the error of b and c two-phase currents comes from the β-axis current, it can be found that the distribution of b and c two-phase current error is almost the same as that of β-axis current error, but the two-phase current error is smaller than the β-axis current error as a whole. The current error is 3.6 mA before entering the stable stage, and the current error is only 2.6 mA after stabilization. This is because the β-axis current coefficient in the Clark transform is less than 1.

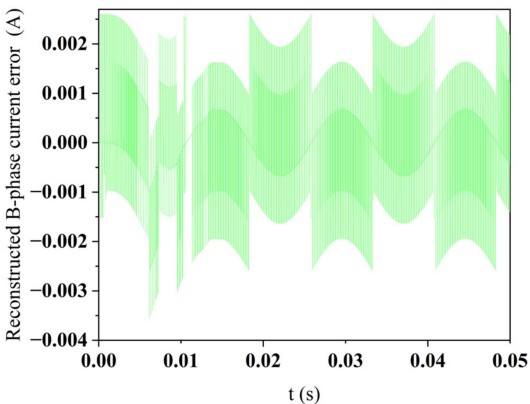

**Figure 4.** Reconstructed B-phase current error.

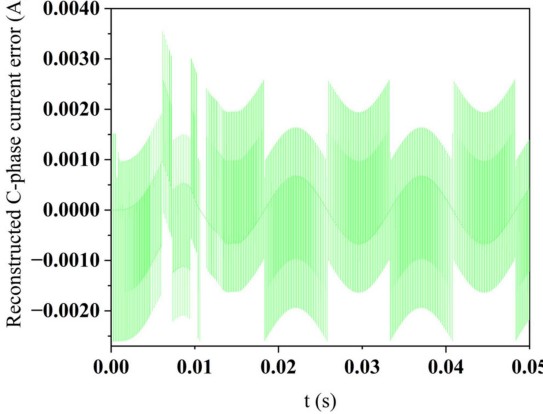

**Figure 5.** Reconstructed C-phase current error.

After applying the reconstructed current to the motor operation, the motor speed and torque shown in Figure 6 can be obtained. The motor speed overshoot of the control system in Figure 1 is only 50 rpm, and the speed tends to be stable at 16 ms. When it is stable, the difference from the set speed is within 1 rpm, and the maximum starting torque is 22 N·m. After the speed is stabilized at 16 ms, the torque is also stabilized and finally fluctuates around 0 N·m positive and negative 0.5 N·m.

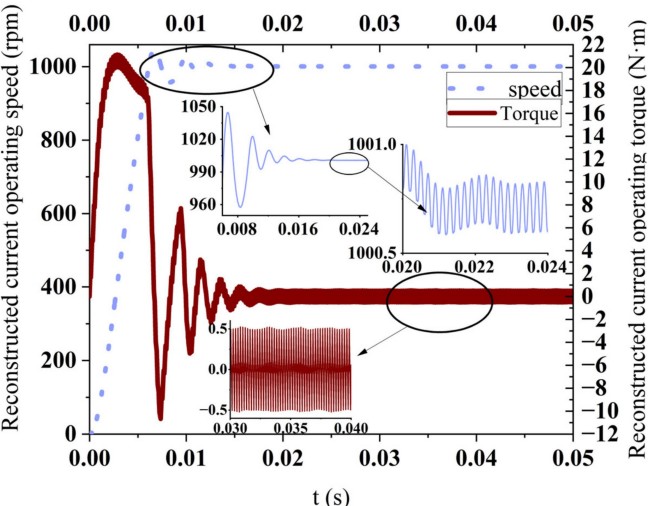

**Figure 6.** The steady-state operating conditions of the reconstructed current.

Based on the above simulation results, it can be confirmed that the current reconstruction system in Figure 1 can achieve fast, accurate, and stable control of the motor under 1000 rpm in a no-load environment. In order to further measure the operation status of the system under the dynamic condition of the motor, the dynamic condition analysis is carried out below. Now assume a complex working condition M: The load at 0 s is 2 N·m, and the speed is set to 600 rpm. The load increases to 5 N·m in 0.02 s, and the speed increases to 1000 rpm. In 0.07 s, the load is reduced to 2 N·m and the speed is reduced to 800 rpm. The speed curve under dynamic working condition m is shown in Figure 7, in the three different set speed changes, the system overshoot is below 50 rpm, and the response time is not prolonged compared with the steady-state condition, but the steady speed becomes larger due to the existence of the load and the deviation of the set speed. In the case of 1000 rpm and 5 N·m, the speed is always 5 rpm different from the set speed. Although the load will increase the deviation between the motor speed and the set speed, it can achieve high performance control of the motor.

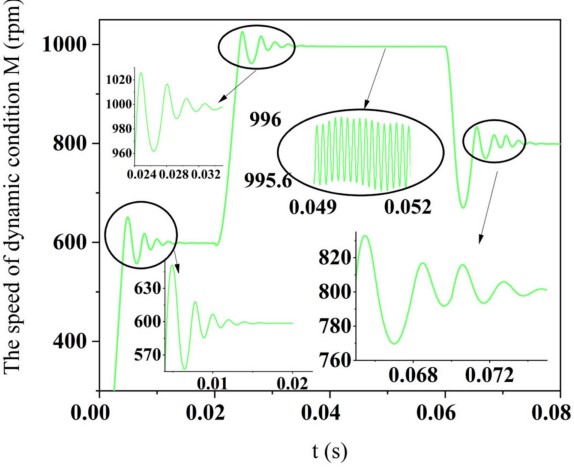

**Figure 7.** The speed of dynamic condition M.

Figure 8 is the torque diagram under operating condition M. In order to verify the current reconstruction ability of the reconstructed current under extremely complex conditions, the speed is changed while the load is changed, this is already a complicated working condition, and the designed system can control the torque from a large fluctuation state to a stable working state in an instant, which fully shows the real-time performance of the system.

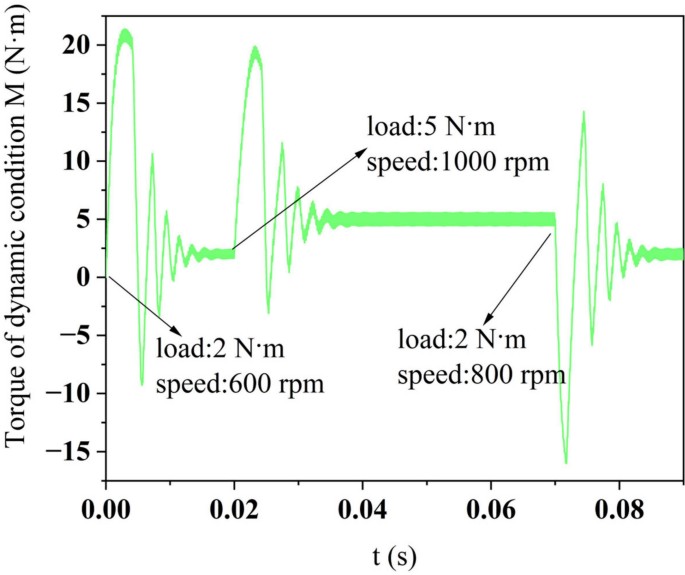

**Figure 8.** Torque of dynamic condition M.

The reconstructed β-axis current under the M condition is shown in Figure 9. Due to the simultaneous jump of speed and load, the current changes at the three time nodes of 0, 0.02, and 0.07 are more severe, but the reconstructed β-axis current can still quickly track the actual β-axis current. The error between the reconstructed β-axis current and the actual β-axis current under dynamic conditions is shown in Figure 10. The maximum error of the reconstructed current is still less than 4 mA, and the position where the maximum current error occurs is the position where the set speed changes. When the motor runs at 1000 rpm and 5 N·m, the current error of 4 mA only accounts for 0.08% of the steady-state current amplitude. Compared with the steady-state, no-load environment, the current error is further reduced.

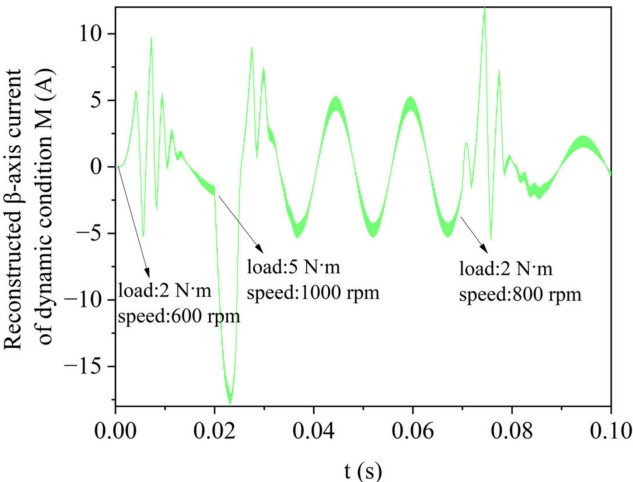

**Figure 9.** Reconstructed β-axis current of dynamic condition M.

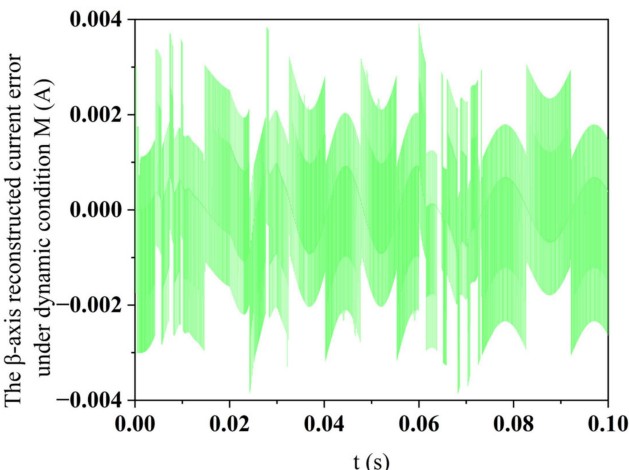

**Figure 10.** The β-axis reconstructed the current error under dynamic condition M.

According to the above analysis, the position of the maximum error of 4 mA of the β-axis reconstruction current appears in two cases: one is the stage where the motor load remains unchanged and the speed rises, as shown in Figure 2; the second is the stage of motor load change and speed change, as shown in Figure 10. According to the above phenomena, it can be judged that the change in motor speed will lead to a sudden change in the reconstructed current error, and then the maximum current error will occur, but whether the load change will lead to the maximum error has not been determined. To this end, increase the working condition N: set the speed at 1000 rpm unchanged, 5 N·m load start, 0.05 s into 15 N·m load.

Figure 11 is the β current reconstruction error under the operating condition N. Under this operating condition, the load of 15 N·m causes the speed to fluctuate at 15 rad at the set speed. The maximum error of the β-axis current is still 4 mA and only occurs once, which proves that the maximum error in the above analysis only appears in the speed adjustment stage. The increase in load will affect the error in the stable stage, such as the spike error in Figure 10. As the load increases, the spike error accumulates into a sharp angle error, but it still does not exceed 4 mA. After further verification, in the normal operating range of the motor, the adjustment of the speed leads to an increase in the local current error, and the introduction of the load leads to an increase in the current error in the stable phase, but the reconstructed current error under the parameter adaptation will not exceed 4 mA.

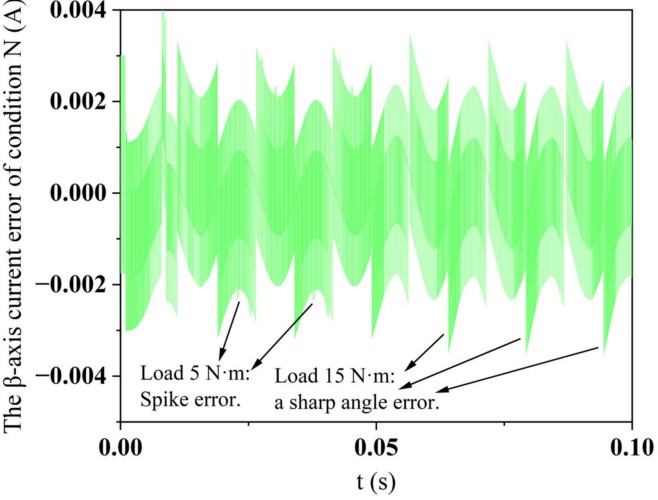

**Figure 11.** The β-axis current error of condition N.

In order to better demonstrate the ability of current reconstruction, the maximum current error in each stage of each working condition accounts for the amplitude of the stable current in this stage, as shown in Table 3. Under various working conditions, the maximum proportion of current error is less than 1%. When the load is added, the proportion of current error is even smaller due to the increase in stable current amplitude, and the lowest proportion is only 0.073%. The proportion of maximum current error reflects local error, which shows that the average error of current will be smaller, and the smaller the current error, the more feasible it is to replace the sensor current value. Table 3 shows the strong current reconstruction ability under steady and dynamic conditions, and the single current sensor FOC control is completely feasible.

**Table 3.** Brief Comparison of Reconstructed Current Error.

| Working Condition | Process | Percentage of Maximum Current Error | Purpose of Working Condition |
|---|---|---|---|
| W | 1000 rpm without load | 0.85% | Steady-state condition verification |
| M | 0 s: 600 rpm with 2 N·m<br>0.02 s: 1000 rpm with 5 N·m<br>0.07 s: 800 rpm with 2 N·m | 0.32%<br>0.073%<br>0.17% | Dynamic condition verification |
| N | 0 s: 1000 rpm with 5 N·m<br>0.05 s: 1000 rpm with 2 N·m | 0.076%<br>0.13% | Locate the position where the maximum error of the current occurs. |

### 4.2. Robustness Verification of Reconstructed Current

In the actual operation of PMSM, due to factors such as temperature and magnetic saturation, parameters such as resistance and inductance will be perturbed, and the performance of the motor control system with poor robustness will be affected. Under the complex working condition M, in order to verify the robustness of the current observer to the resistance, inductance, and flux linkage, the parameter perturbation is introduced for analysis.

Figure 12 is the $\beta$-axis current reconstruction error of $\hat{R}_S = 1.3R_S$, M and M without load. The speed changes at three moments of 0, 0.02, and 0.07, resulting in a large current error, but it quickly converges to $\pm 1$ A, and the no-load current error only fluctuates within 5 mA. Although the current error will increase to 1 A when the load is 5 N·m in 0.02 s, the average current error is 0, and the motor can still operate at high performance, which indicates that the current observer has good robustness to the stator resistance.

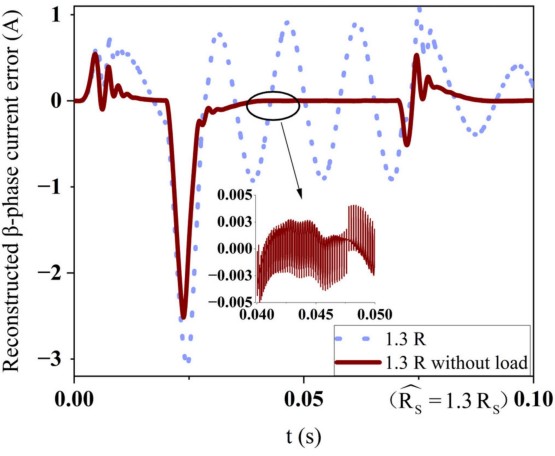

**Figure 12.** Mismatched $\beta$-axis current error.

The mismatch of $L_d$ will also affect the operating performance of the motor. The $\beta$-axis current reconstruction error under M condition and M condition without load is

shown in Figure 13. Compared with the current reconstruction performance when the stator resistance $R_S$ is mismatched, the mismatch of $L_d$ causes the current error to fluctuate violently when the speed is adjusted, but the current error can still be stabilized quickly. Even at 1000 rpm and a 5 N·m load, the stable current error peak does not exceed 1 A. Obviously, the current observer has good robustness for the stator inductance.

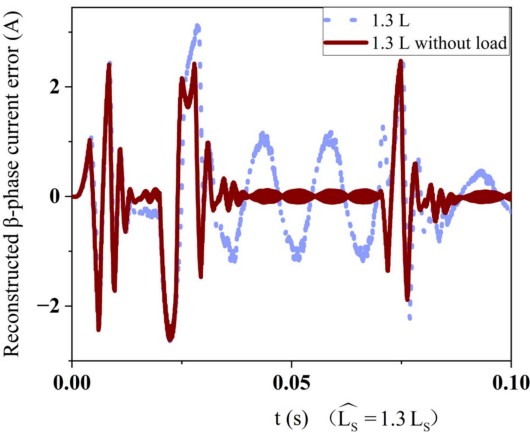

**Figure 13.** Mismatched β-axis current error.

Figure 14 still shows the β-axis current reconstruction error of M condition and M condition without load at this time $\hat{\psi}_f = 1.05\ \psi_f$. Different from the inductance and resistance mismatch, the applied load has little effect on the reconstructed current error, and the peak current error is below 0.9 A. However, the degree of flux mismatch is much smaller than the resistance and inductance in Figures 12 and 13. Obviously, the flux mismatch has a greater impact on the current error. When $\hat{\psi}_f = 1.5\psi_f$, the no-load double current sensor speed of M condition, the double current sensor speed of M condition, and the single current sensor speed of M condition are shown in Figure 15. When controlled by double current sensors, the fluctuation of no-load speed is equivalent to that of loaded speed controlled by a single current sensor. However, when controlled by double current sensors with load, speed fluctuation is much higher than when controlled by a single current sensor with load. At 1000 rpm, the speed fluctuation of a dual-current sensor with load control is up to 40 rpm, and the fluctuation of a dual-sensor no-load and a single-sensor load control is only 10 rpm. Even in the case of parameter mismatch, the reconstructed current can achieve the actual current control effect. So far, the current observer has shown excellent robustness to motor parameters.

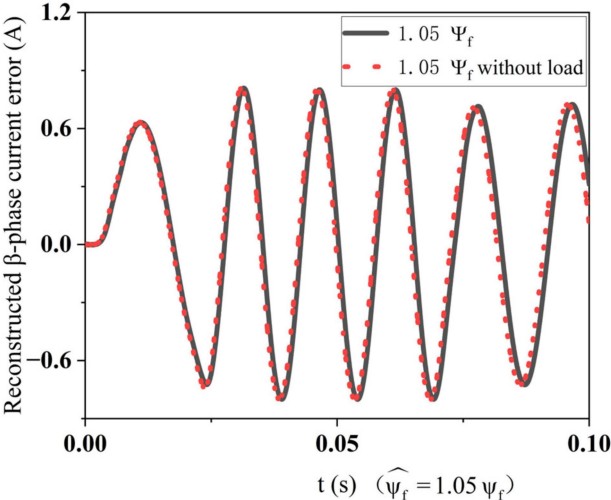

**Figure 14.** Mismatched β-axis current error.

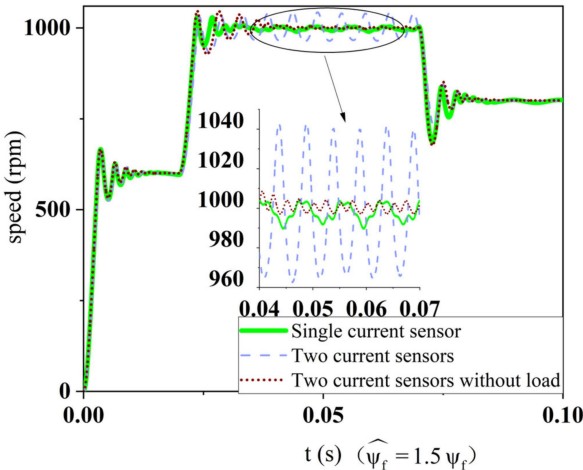

**Figure 15.** Mismatched speed.

### 4.3. Comparison of Control Effect between Reconstructed Current and Actual Current

This design is designed to replace PMSM multi-current sensor control. In order to verify that the reconstructed current can replace the actual current, the effects of FOC dual-current sensor control, V/f control, and improved direct torque control (IDTC) are compared. The control effects are shown in Figure 16, in which Figure 16a shows the speed curve and Figure 16b shows the torque curve. The dynamic performance of V/f control alone is far worse than that of other control methods, and the response time is relatively long, so it can be considered to be combined with other algorithms in special circumstances. IDTC has a fast dynamic response and a good control effect, but the speed and torque overshoot are large. The control effect of the FOC single current sensor reaches the control effect of the actual current even better because the error of the current observer is small, which is not obvious compared with the control effect of the FOC with position sensor. The FOC double current sensor used here has no position sensor. Generally speaking, the more mature control method of reducing sensors is not to use the position sensor, and this control effect comparison diagram shows that in the FOC control method, the single current sensor is better than the double current sensor without the position sensor to some extent. When the effect of position-free control using a single current sensor is not ideal, a single current sensor with a position sensor can be selected for control, which can reduce the number of sensors while still achieving high performance control.

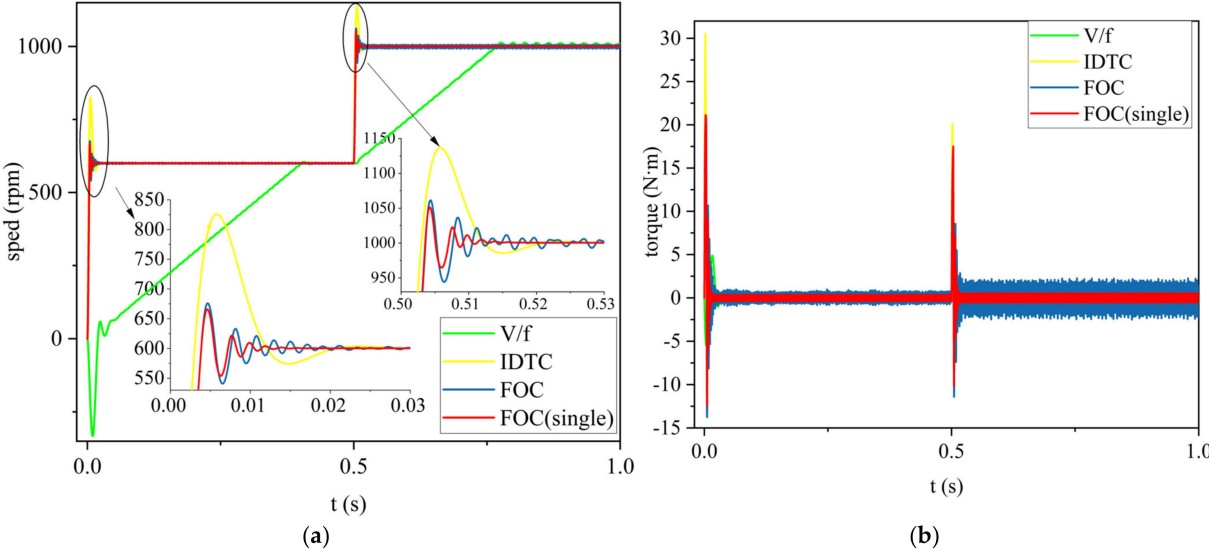

**Figure 16.** The performance of PMSM under four control methods. (**a**) Speed; (**b**) Torque.

### 5. Conclusions

After analyzing the current acquisition scheme and the current equation of the motor, a sliding mode current observer is proposed to realize the single current sensor control of the PMSM. Under steady-state conditions, the maximum error of the reconstructed current is less than 4 mA, accounting for only 0.85% of the steady-state current amplitude. Under the dynamic condition, the speed and load change suddenly at the same time, and the current tracking ability is still very strong. Due to the addition of the load, the stable current rises, accounting for only 0.17% of the steady-state current amplitude. In the robustness test, the mismatch of inductance, resistance, and flux linkage leads to an increase in the local error of the reconstructed current, which can reach 3 A, but the average error fluctuates around 0 A, and the motor can still maintain high performance control. Compared with V/f, DTC, and FOC, the single current FOC algorithm has low overshoot, a fast dynamic response, and stable operation. Through a series of tests and analyses, the following conclusions can be drawn:

(1) There is no connection between the designed current observer and the DC bus method. There is no blind area in the current reconstruction. There is no need to obtain other current information except the a-phase current, and no additional hardware measures are required.

(2) The current observer does not introduce complex structure. It has simple calculation, low dependence on parameters, high robustness, and high precision in the current reconstruction. It can achieve high-performance control of the motor.

(3) The disadvantage of this design is that a position sensor is needed. Further research considers the current reconstruction without a position sensor, but the position acquisition does not consider the use of observers. Multiple observers will increase the complexity of the system. This design can replace position sensorless FOC control. FOC without a position sensor generally adopts double current sensors and reduces one position sensor at the same time, but the current observer designed in this paper can reduce one current sensor. The current sensor can achieve better control in the case of reducing one current sensor, as shown in Figure 16.

In addition, in the error analysis, we found that the maximum local error occurs in the speed adjustment stage, which is independent of the load; however, what is related to the load is the stab error in the stable stage. As the load increases, the spike error will accumulate into a sharp angle error. Due to the small current error, the local maximum error is not much different from the spike error, but this finding may be of great significance in trying to further apply the current observer.

**Author Contributions:** Conceptualization, G.Y. and Z.W.; methodology, Y.X.; software, Y.Y.; validation, G.Y., Z.W. and Y.X.; formal analysis, Y.Y.; investigation, Y.Y.; resources, Y.Y.; data curation, Y.Y.; writing—original draft preparation, Y.Y.; writing—review and editing, G.Y.; visualization, Y.Y.; supervision, Y.X.; project administration, G.Y.; funding acquisition, G.Y. All authors have read and agreed to the published version of the manuscript.

**Funding:** This work is supported in part by the National Natural Science Foundation of China under Grant 52066008, the Development of domestic Electronic Control System (ECU) for the China VI diesel engine under Grant 202104BN050007, the Key Technology Research and Development of a methanol/diesel dual fuel engine under Grant 202103AA080002, and the Research and Application of Key Technologies for extended-range commercial electric vehicles under Grant 202102AC080004.

**Data Availability Statement:** Data are available upon request from the authors.

**Conflicts of Interest:** The authors declare no conflict of interest.

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
