# Peer review of "Permanent Magnet Synchronous Motor Control Based on Phase Current Reconstruction"

_electronics, doi:10.3390/electronics12071624_

Round 1
Reviewer 1 Report
The topic of the paper is interesting and up to date, however some aspects need to be improved:
- in the introduction it should be better specified what is the research gap that we intend to fill with the following study;
- in the introduction the innovative points of the article with respect to the present literature should be highlighted in a bullet list;
- in the introduction highlight other studies that consider the uncertainty study of torque measurement and number of revolutions of PMSM, such as:
De Santis, M.; Agnelli, S.; Patanè, F.; Giannini, O.; Bella, G. Experimental Study for the Assessment of the Measurement Uncertainty Associated with Electric Powertrain Efficiency Using the Back-to-Back Direct Method. Energies 2018, 11, 3536. https://doi.org/10.3390/en1112353.
- In the model that is described initially, why are only 2 phases A and B and not three phases A, B and C given as input to the module that does the transform into alpha and beta (Clarke's transform), as is then reported in the results?
- From the described observer method it is not clear if it is a new observer model that is proposed or if it is a known model used for this case study?
- in the results the mismatch that leads to possible overshoots of the measured quantities, to which type of mismatch does it refer? please specify.
- in the final part of the results, in addition to showing the performance differences using measurements with double sensor or single sensor, is it possible to consider comparisons with other torque and speed control techniques?
- Is it possible to provide a table with the percentage errors of the various case studies considered, in order to clarify the performance of the observer with a single current sensor compared to one with two sensors?
Author Response
Dear Editor:
Thank you very much for your careful guidance.Please do not hesitate to contact me if any further information is needed. Here are our answers.For a better reading experience, you can download the attachment for viewing.
Reviewer #1:
The topic of the paper is interesting and up to date, however some aspects need to be improved:
Reviewer’s question/comment 1:
in the introduction it should be better specified what is the research gap that we intend to fill with the following study;
Authors’ reply 1:
Thank you very much for your great feedback. I rewrote the last paragraph of the introduction to replace the original.
The following contents have been added in the revised manuscript on page 2 and lines 108-116, copied below for your convenience:
At present, regarding the PMSM single current sensor control system, there is no system that can completely eliminate the blind area of current reconstruction, and has the characteristics of simple structure and strong robustness. The elimination of the blind area of current reconstruction will inevitably make the motor run more smoothly. Without introducing complex structure, the anti-interference ability is greatly enhanced, the real-time control effect is better, and the motor can adapt to more complex working environment. Therefore, the purpose of this paper is to realize the current re-construction with no blind area, high precision and good robustness by a very simple method, and then realize the high performance control of PMSM by a single current sensor.
Reviewer’s question/comment 2:
in the introduction the innovative points of the article with respect to the present literature should be highlighted in a bullet list;
Authors’ reply 2:
We are sorry for not making a clear description about the innovation points. We try to add a table to briefly describe this.
Table 1. Brief comparison of single current control strategies
|
Method |
Reference |
Advantages |
Disadvantages |
|
DC bus method
|
Reference [19] |
The DC bus method is proposed. |
The current blind area. |
|
Reference [20] |
Reduces the blind area and the common-mode voltage. |
not as good as the traditional 7-segmentSVPWM. |
|
|
Reference [21] |
Need an isolated current sensor. |
Additional lead wire. |
|
|
Reference [22] |
PWM Phase Shift. |
cannot eliminate all blind area. |
|
|
Reference [23] |
One Branch. |
A new blind area. |
|
|
Reference [24] |
Switching state phase shift method. |
Current ripple and switch damage. |
|
|
Reference [25] |
Substitution method. |
Ignore the sector boundary. |
|
|
Current observer method |
Reference [26] |
Observer equations and adaptive laws are not complicated. |
Robustness depends on parameter identification. |
|
Reference [27] |
Accurate. |
ESO parameters. |
|
|
Reference [28] |
Based on PI loop. |
PI ring is not robust. |
|
|
Reference [29] |
EKF. |
Low accuracy. |
|
|
Reference [30] |
The estimation technique is independent of machine parameters. |
The robustness of stator resistance is not very good. |
|
|
|
Reference [31] |
The DC bus method and observer method are combined. |
A blind area in the low modulation region. |
Reviewer’s question/comment 3:
in the introduction highlight other studies that consider the uncertainty study of torque measurement and number of revolutions of PMSM, such as:
De Santis, M.; Agnelli, S.; Patanè, F.; Giannini, O.; Bella, G. Experimental Study for the Assessment of the Measurement Uncertainty Associated with Electric Powertrain Efficiency Using the Back-to-Back Direct Method. Energies 2018, 11, 3536. https://doi.org/10.3390/en1112353.
Authors’ reply 3:
We are sorry that in the introduction highlight other studies. It has been revised, and 13 literatures about current reconstruction are discussed, with other literatures as background supplements.
As early as 1989, the DC bus method has been proposed [19], its essence is to use the instantaneous current of the inverter fixed point contains multi-phase current information to reconstruct the current. However, it takes time to collect current information, and the effective voltage vector has a short action time, which leads to the existence of current reconstruction blind area. In recent decades, many improvement strategies have been proposed to overcome the current blind spot. Reference [20] pro-posed a three-state pulse width modulation technology, which divides the space vector into two regions. In the low modulation ratio region, two basic voltages with a difference of 120 ° are used to synthesize the reference voltage. This technology reduces the blind area of current reconstruction and suppresses the common mode voltage, but its performance is not as good as the traditional seven-segment space vector pulse width modulation control method. In reference [21], it is proposed to use an isolated current sensor instead of a DC bus current sensor to sample twice in each PWM cycle. However, this method requires additional leads, which will introduce parasitic parameters and lead to system performance degradation. At the same time, the current reconstruction blind area in the overmodulation region is too large, which is also an important factor limiting the method. The PWM phase shift method proposed in Reference [22] can increase the observable area of the current by moving the PWM wave-form when the non-zero voltage vector action time is too short, but this method still has the voltage vector action time after phase shift less than the minimum sampling time. Reference [23] proposed a new phase current reconstruction technique, in which a single current sensor is installed on a branch rather than a DC bus. The blind area of current reconstruction in the sector boundary region and the low modulation region can be reduced without introducing any additional algorithm compensation strategy, but there is a blind area in the high modulation ratio region. Reference [24] proposed a new phase current reconstruction scheme without using zero-switching state. The current reconstruction dead zone is divided into six sectors. Each sector is divided into three parts by the corresponding vector synthesis method to obtain enough effective switching states, thereby reducing the current reconstruction dead zone, but it will increase the current ripple and switching loss and reduce the service life of the system. Reference [25] proposed an idea of " substitution. " When the reference voltage vector is located in the blind area of overmodulation reconstruction, it is replaced by the adjacent reconfigurable maximum voltage vector to broaden the operating range of the phase current reconstruction technology, but this does not solve the blind area problem in other areas.
The core idea of the above improvement measures is to be able to collect the re-quired current information in time by changing the space voltage vector or switching state in a limited time. However, due to factors such as high dynamic response of motor control, the requirements of space voltage vector and switching are very high. Once these measures are adopted, it is not only difficult to completely eliminate the blind area of current reconstruction, but also has a negative impact on the control performance of the motor. Therefore, the current observer becomes another direction of current reconstruction. The current observer method without low modulation ratio region and sector transition region has certain advantages.
In [26], an adaptive observer is proposed to realize the single current control of the motor. The observer equation and adaptive law are not complicated, but its robustness depends on parameter identification. The error of parameter identification may also affect the current reconstruction. The ESO observer designed in Reference [27] can realize the accurate control of single current sensor, but the parameter design of ESO is very difficult. It is usually obtained by trial and error method, which not only increases the workload of the algorithm, but also makes it difficult to obtain the optimal parameters. Verma et al. [28] reconstructed the β-axis current based on the mathematical model of the motor and the PI loop, but the anti-interference ability of the PI loop needs to be improved. Reference [29] designed a single current sensor control algorithm based on Kalman filter, but the calculation of the correction current is not accurate enough, the control performance is poor, the motor torque accuracy is low, and there is a non-periodic pulse spike. Reference [30] can realize single current sensor control without position sensor, but the robustness of the system to the stator resistance is not very good. Reference [31] combines the DC bus method and the observer method, and uses the Luenberger observer to compensate for the limitation of the sector boundary region, but there is still a blind area in the low modulation region.
After changing the manuscript, I want to share the quoted ideas in the introduction of the literature.
In the four references cited at the beginning of this paper, although other studies are described, we believe that a variety of studies on PMSM are precisely the performance of PMSM receiving widespread attention.
Permanent magnet synchronous motor (PMSM) has the advantages of high power density and fast dynamic response. It is widely used in industrial production, new energy vehicles, ship propulsion and other fields [1-4].
Reference 1. Permanent Magnet Synchronous Machine Torque Estimation Using Low Cost Hall-Effect Sensors.
“Permanent magnet synchronous machines (PMSMs) are commonly used in industrial automation for traction, robotics, wind energy, or aerospace applications.”
Reference 2. New Fuzzy Speed Controller For Dual Star Permanent Magnet Synchronous Motor.
“Over last years, electric machines gained more attention in many fields, they are intensively used in electric ships propulsion, electric and more electric aircrafts, electric and hybrid vehicles and especially in renewable energy systems such as wind and tide”
Reference 3. Robust Fault Detection for Permanent-Magnet Synchronous Motor via Adaptive Sliding Mode Observer.
“With its high efficiency, low energy consumption, light-weight, low noise, maintainability, and many other advantages, the permanent-magnet synchronous motor(PMSM) has absorbed strong interest from many scholars. For the rapid development of rail transit, new energy vehicles, aviation, ships, and other fields.
Reference 4. Research and Analysis of Permanent Magnet Transmission System Controls on Diesel Railway Vehicles.”
“As the energy crisis and environmental pollution continue to be a gradual threat, the energy saving of transmission systems has become the focus of railway vehicle research and design. Due to their high-power density and efficiency features, permanent magnet synchronous motors (PMSM) have been gradually applied in railway vehicles.”
Firstly, it is described that PMSM is very concerned [1-4]. Then, when it comes to his control strategy, it is necessary to introduce the unadopted V/f [5-7] and DTC [8-10], and state the reasons for abandoning them, and describe some advantages of FOC [12-14]. After affirming the advantages of FOC, a single sensor is proposed for current feedback link selection [15-18]. Then two main methods of single current sensor strategy are analyzed [19-24]. What I want to share is the process of thinking, and make a diagram-assisted explanation on the next.
Thank you very much for your reminder, let me realize that other studies occupy too much space.
Reviewer’s question/comment 4:
In the model that is described initially, why are only 2 phases A and B and not three phases A, B and C given as input to the module that does the transform into alpha and beta (Clarke's transform), as is then reported in the results?
Authors’ reply 4:
The FOC algorithm requires three-phase current to control. However, due to Kirchhoff 's law, only two current sensors are needed. It is customary to measure the A-phase current and the B-phase current to obtain the three-phase current. What this paper needs is to point out that the α-axis current is only related to the A-phase current, and the β-axis current can be calculated only by the A-phase current and the B-phase current, so as to serve the subsequent observer design, so the C-phase current is replaced by the other two phases. The specific derivation process is as follows:
Clarke's transform:
|
|
(1) |
|
Under 3-phase balanced current condition: |
|
|
(2) |
|
|
(3) |
Substitute C phase current:
|
|
(4) |
|
(5) |
According to equation (5), it is obvious that as long as the A phase current is measured, the correct α-axis current can be obtained, while if the B phase current is measured, neither α-axis current nor β-axis current can be obtained, which is an important idea in the design of current observer.
Reviewer’s question/comment 5:
From the described observer method it is not clear if it is a new observer model that is proposed or if it is a known model used for this case study。
Authors’ reply 5:
In the single current sensor FOC algorithm with position sensor, I think this is a new observer. However, for the synovial observer theory, this is not enough to be considered as a new synovial observer. The design of synovial convergence and synovial surface is based on the classical synovial theory. We cannot regard the observer with new synovial surface and some small changes as a new synovial observer. However, there is no doubt that this observer has a very good performance in the current reconstruction of the position sensor. When we consider reducing a sensor to carry out the motor sensor, in addition to the choice of removing the position sensor, we can also consider the scheme of this paper. Under the same conditions, the control performance of this scheme is much better than that of sensorless.
Reviewer’s question/comment 6:
in the results the mismatch that leads to possible overshoots of the measured quantities, to which type of mismatch does it refer? please specify.
Authors’ reply 6:
The parameters of a motor should be fixed, but the operating environment of the motor is complex, and the parameters often change when the motor is running, which is no longer in line with the motor control system. In this paper, the mathematical model equation of the motor is used, and the notable parameters are mainly stator resistance, stator inductance and stator flux linkage. These three parameters have great influence on the whole control system, and once they exceed a certain degree, every parameter will become irregular, the control system will collapse. The system designed in this paper has the worst robustness to stator flux linkage. Let's take stator flux linkage mismatch as an example.
The stator current is both the input and the output of the system, and the observed current is impacted by the mismatch parameters earlier. The following figure shows the actual current and reconstructed current diagram of stator flux mismatch. It is worth noting that stator resistance and inductance mismatch will also have the same negative impact, but the system is more robust to inductance and resistance and will be less damaged.
Figure 1. Actual current and reconstructed current when stator flux linkage becomes 1.5 times.
Figure 2. Reconstructed current error when stator flux linkage becomes 1.5 times.
Figure 3. Speed when stator flux linkage becomes 1.5 times.
Figure 4. Torque when stator flux linkage becomes 1.5 times.
It can be found that the mismatch of flux linkage will affect the control performance of the motor system, rather than just one parameter affecting individual parameters. The mismatch of resistance and inductance will also affect the whole system, and it will not be shown because of the limited space. When the stator flux linkage is 1.5 times of the original value, the average error of the reconstructed current is not great because of the good robustness of the system, and the system speed can still be maintained near the set speed. To make an extreme verification, when the stator flux linkage becomes 6 times of the original, the motor control system will lose control, as shown in the following figure. When the stator flux linkage is 6 times of the original, the tracking ability of the current observer is basically lost, the current error is increased and irregular, the torque fluctuation is increased, and the speed is completely out of control. The mismatch of any parameter of resistance, inductance and flux linkage in this design will have a negative impact on the whole system, the current tracking ability will decrease, the deviation between speed and set speed will increase, and the torque fluctuation will increase.
Figure 5. Actual current and reconstructed current when stator flux linkage becomes 6 times.
Figure 6. Reconstructed current error when stator flux linkage becomes 6 times.
Figure 7. Speed when stator flux linkage becomes 6 times.
Figure 8. Torque when stator flux linkage becomes 6 times.
Reviewer’s question/comment 7:
in the final part of the results, in addition to showing the performance differences using measurements with double sensor or single sensor, is it possible to consider comparisons with other torque and speed control techniques?
Authors’ reply 7:
Thank you for your valuable comments. I will add a comparison of FOC (including single sensor and two current sensors), V/f and DTC control.
The following contents have been added in the revised manuscript on page 2 and lines 376-395, copied below for your convenience:
4.3 Comparison of Control Effect between Reconstructed Current and Actual Current
This design is designed to replace PMSM multi-current sensor control. In order to verify that the reconstructed current can replace the actual current, the effects of FOC dual-current sensor control, V/f control and improved direct torque control (IDTC) are compared. The control effects are shown in Figure 16, in which Figure a shows the speed curve and Figure b shows the torque curve. The dynamic performance of V/f control alone is far worse than other control methods, and the response time is relatively long, so it can be considered to be combined with other algorithms in special occasions. IDTC has fast dynamic response and good control effect, but the speed and torque overshoot are large. The control effect of FOC single current sensor reaches the control effect of actual current, even better, because the error of current observer is small, which is not obvious compared with the control effect of FOC with position sensor. The FOC double current sensor used here has no position sensor. Generally speaking, the more mature control method of reducing sensors is not to use the position sensor, and this control effect comparison diagram shows that in the FOC control method, the single current sensor is better than the double current sensor without the position sensor to some extent. When the effect of position-free control using single current sensor is not ideal, single current sensor with position sensor can be selected for control, which can reduce the number of sensors and still achieve high performance control.
|
(a) |
(b) |
Figure 16. The performance of PMSM under four control methods. (a)speed; (b)torque.
Reviewer’s question/comment 8:
Is it possible to provide a table with the percentage errors of the various case studies considered, in order to clarify the performance of the observer with a single current sensor compared to one with two sensors?
Authors’ reply 8:
Since the current error comes from the sensor and the measuring environment itself when the dual-current sensor scheme is adopted, and the three-phase current is directly available, the control strategy considers that the dual-current sensor has no current error. When the single-current sensor strategy can achieve the effect of dual-current sensor, the reconstruction current can be considered to be successful to a certain extent. I will add a table to describe the error of single current sensor under some working conditions. In the case of robustness verification, due to the large fluctuation of current error, the average error has more research significance. However, in the case of general parameter mismatch, the average error of the system with good robustness fluctuates at the 0 attachment, so it is not listed in the table.
The following contents have been added in the revised manuscript on page 2 and lines 225-229, copied below for your convenience:
In order to better demonstrate the ability of current reconstruction, the maximum current error in each stage of each working condition accounts for the amplitude of the stable current in this stage, as shown in Table 3. Under various working conditions, the maximum proportion of current error is less than 1%. When the load is added, the proportion of current error is even smaller due to the increase of stable current amplitude, and the lowest proportion is only 0.073%. The proportion of maximum current error reflects local error, which shows that the average error of current will be smaller, and the smaller the current error, the more feasible it is to replace the sensor current value. Table 3 shows the strong current reconstruction ability under steady and dynamic conditions, and the single current sensor FOC control is completely feasible.
Table 3. Brief Comparison of Reconstructed current error.
|
Working Condition |
Process |
Percentage of Maximum Current Error |
Purpose of Working Condition |
|
W |
1000rpm without load |
0.85% |
Steady-state condition verification |
|
M |
0s: 600rpm with 2N·m |
0.32% |
Dynamic condition verification |
|
0.02s: 1000rpm with 5N·m |
0.073% |
||
|
0.07s: 800rpm with 2N·m |
0.17% |
||
|
N |
0s: 1000rpm with 5N·m |
0.076% |
Locate the position where the maximum error of current occurs. |
|
0.05s: 1000rpm with 2N·m |
0.13% |

Reviewer 2 Report
The paper presents aspects related to the permanent magnet synchronous motor control based on phase current reconstruction and is useful to electrical shareholders. The control system tests the performance of the motor under different working conditions when the parameters are 15 matched, and then tests the current reconstruction ability of the parameter mismatch.
The subject is not new but the approach is interesting and useful for specialists in the field. The large amount of data, graphs, representations and discussions based on the data presented represent the strong and new part brought by the paper. The methodology used is good and allows to highlight the problems discussed. The tables and figures is good.
Suggestions on revision:
The main problem is related to the formulation of the conclusions that must summarize the advantages, disadvantages and main problems from the studies.
The conclusions are incomplete and do not reflect the content of the data presented in the paper, which must be reformulated and improved. They must result from the studies done and the processed data.
The references are at the limit can be improved.
Author Response
Dear sir or madam:
Thank you very much for your careful guidance in your busy schedule, and we feel honored for it.We have reflected on your guidance, and the details are contained in the annex. For a better reading experience, you can download the attachment for viewing.Have a nice day!
Reviewer’s question/comment 1:
The main problem is related to the formulation of the conclusions that must summarize the advantages, disadvantages and main problems from the studies.
Authors’ reply 1:
Thank you very much for your great feedback. After my reflection, I summarized some advantages and disadvantages.
The following contents have been added in the revised manuscript on page 2 and lines 408-430, copied below for your convenience:
Through a series of tests and analysis, the following conclusions can be drawn:
(1) There is no connection between the designed current observer and the DC bus method. There is no blind area of current reconstruction. There is no need to obtain other current information except the a-phase current, and no additional hardware measures are required.
(2) The designed current observer does not introduce complex structure. It has simple calculation, low dependence on parameters, high robustness and high precision of current reconstruction. It can achieve high performance control of the motor.
(3) The disadvantage of this design is that the position sensor is needed. Further research considers the current reconstruction without position sensor, but the position acquisition does not consider the use of observers. Multiple observers will increase the complexity of the system. This design can be applied to the position sensorless alternative. FOC without position sensor generally adopts double current sensors and reduces one position sensor at the same time, but the current observer designed in this paper can reduce one current sensor. The current sensor can achieve better control in the case of reducing one sensor, as shown in Figure 16.
In addition, in the error analysis, it is found that the maximum local error occurs in the speed adjustment stage, which is independent of the load; what is related to the load is the spike error in the stable stage. As the load increases, the spike error will ac-cumulate into a sharp angle error. Due to the small current error, the local maximum error is not much different from the spike error, but this finding may be of great significance in trying to further apply the current observer.
Reviewer’s question/comment 2:
The conclusions are incomplete and do not reflect the content of the data presented in the paper, which must be reformulated and improved. They must result from the studies done and the processed data.
Authors’ reply 2:
Thank you very much for your kind guidance I rewrote the conclusion, which contains the advantages and disadvantages just mentioned.
After analyzing the current acquisition scheme and the current equation of the motor, a sliding mode current observer is proposed to realize the single current sensor control of PMSM. Under steady-state conditions, the maximum error of the reconstructed current is less than 4mA, accounting for only 0.85 % of the steady-state current amplitude. Under the dynamic condition, the speed and load change suddenly at the same time, and the current tracking ability is still very strong. Due to the addition of the load, the stable current rises, accounting for only 0.17 % of the steady-state current amplitude. In the robustness test, the mismatch of inductance, resistance and flux linkage leads to the increase of local error of reconstructed current, which can reach 3A, but the average error fluctuates around 0A, and the motor can still maintain high performance control. Compared with V/f, DTC and FOC, the single current FOC algorithm has low overshoot, fast dynamic response and stable operation. Through a series of tests and analysis, the following conclusions can be drawn:
(1) There is no connection between the designed current observer and the DC bus method. There is no blind area of current reconstruction. There is no need to obtain other current information except the a-phase current, and no additional hardware measures are required.
(2) The designed current observer does not introduce complex structure. It has simple calculation, low dependence on parameters, high robustness and high precision of current reconstruction. It can achieve high performance control of the motor.
(3) The disadvantage of this design is that the position sensor is needed. Further research considers the current reconstruction without position sensor, but the position acquisition does not consider the use of observers. Multiple observers will increase the complexity of the system. This design can be applied to the position sensorless alternative. FOC without position sensor generally adopts double current sensors and reduces one position sensor at the same time, but the current observer designed in this paper can reduce one current sensor. The current sensor can achieve better control in the case of reducing one sensor, as shown in Figure 16.
In addition, in the error analysis, it is found that the maximum local error occurs in the speed adjustment stage, which is independent of the load; what is related to the load is the spike error in the stable stage. As the load increases, the spike error will ac-cumulate into a sharp angle error. Due to the small current error, the local maximum error is not much different from the spike error, but this finding may be of great significance in trying to further apply the current observer.
Reviewer’s question/comment 3:
The references are at the limit can be improved.
Authors’ reply 3:
Thank you very much for your great feedback. In the introduction part of the current reconstruction method, I deleted some references and added more wonderful references. The total number of references in this part increased from 6 to 13. I want to share the following references cited ideas, hoping that the changes in references are indeed good.
As early as 1989, the DC bus method has been proposed [19], its essence is to use the instantaneous current of the inverter fixed point contains multi-phase current information to reconstruct the current. However, it takes time to collect current information, and the effective voltage vector has a short action time, which leads to the existence of current reconstruction blind area. In recent decades, many improvement strategies have been proposed to overcome the current blind spot. Reference [20] pro-posed a three-state pulse width modulation technology, which divides the space vector into two regions. In the low modulation ratio region, two basic voltages with a difference of 120 ° are used to synthesize the reference voltage. This technology reduces the blind area of current reconstruction and suppresses the common mode voltage, but its performance is not as good as the traditional seven-segment space vector pulse width modulation control method. In reference [21], it is proposed to use an isolated current sensor instead of a DC bus current sensor to sample twice in each PWM cycle. However, this method requires additional leads, which will introduce parasitic parameters and lead to system performance degradation. At the same time, the current reconstruction blind area in the overmodulation region is too large, which is also an important factor limiting the method. The PWM phase shift method proposed in Reference [22] can increase the observable area of the current by moving the PWM wave-form when the non-zero voltage vector action time is too short, but this method still has the voltage vector action time after phase shift less than the minimum sampling time. Reference [23] proposed a new phase current reconstruction technique, in which a single current sensor is installed on a branch rather than a DC bus. The blind area of current reconstruction in the sector boundary region and the low modulation region can be reduced without introducing any additional algorithm compensation strategy, but there is a blind area in the high modulation ratio region. Reference [24] proposed a new phase current reconstruction scheme without using zero-switching state. The current reconstruction dead zone is divided into six sectors. Each sector is divided into three parts by the corresponding vector synthesis method to obtain enough effective switching states, thereby reducing the current reconstruction dead zone, but it will increase the current ripple and switching loss and reduce the service life of the system. Reference [25] proposed an idea of " substitution. " When the reference voltage vector is located in the blind area of overmodulation reconstruction, it is replaced by the adjacent reconfigurable maximum voltage vector to broaden the operating range of the phase current reconstruction technology, but this does not solve the blind area problem in other areas.
The core idea of the above improvement measures is to be able to collect the re-quired current information in time by changing the space voltage vector or switching state in a limited time. However, due to factors such as high dynamic response of motor control, the requirements of space voltage vector and switching are very high. Once these measures are adopted, it is not only difficult to completely eliminate the blind area of current reconstruction, but also has a negative impact on the control performance of the motor. Therefore, the current observer becomes another direction of current reconstruction. The current observer method without low modulation ratio region and sector transition region has certain advantages.
In [26], an adaptive observer is proposed to realize the single current control of the motor. The observer equation and adaptive law are not complicated, but its robustness depends on parameter identification. The error of parameter identification may also affect the current reconstruction. The ESO observer designed in Reference [27] can realize the accurate control of single current sensor, but the parameter design of ESO is very difficult. It is usually obtained by trial and error method, which not only increases the workload of the algorithm, but also makes it difficult to obtain the optimal parameters. Verma et al. [28] reconstructed the β-axis current based on the mathematical model of the motor and the PI loop, but the anti-interference ability of the PI loop needs to be improved. Reference [29] designed a single current sensor control algorithm based on Kalman filter, but the calculation of the correction current is not accurate enough, the control performance is poor, the motor torque accuracy is low, and there is a non-periodic pulse spike. Reference [30] can realize single current sensor control without position sensor, but the robustness of the system to the stator resistance is not very good. Reference [31] combines the DC bus method and the observer method, and uses the Luenberger observer to compensate for the limitation of the sector boundary region, but there is still a blind area in the low modulation region.

Round 2
Reviewer 1 Report
The authors improved the paper according to the reviewer's comments.
The paper is suitable for publication.
Reviewer 2 Report
The paper can be published